

# Estimating food resource availability in arid environments with Sentinel 2 satellite imagery

Caterina Funghi[1,2], René H.J. Heim[2,3], Wiebke Schuett[1,4,5], Simon C. Griffith[2,5] and Jens Oldeland[3]

[1] Institute of Zoology, Universität Hamburg, Hamburg, Germany
[2] Department of Biological Sciences, Macquarie University, Sydney, New South Wales, Australia
[3] Institute for Plant Science and Microbiology, Universität Hamburg, Hamburg, Germany
[4] School of Life Sciences, University of Sussex, Falmer, Brighton, United Kingdom
[5] Department of Biological, Earth and Environmental Sciences, University of New South Wales, Sydney, New South Wales, Australia

Corresponding author
Caterina Funghi,
caterina.funghi@students.mq.edu.au,
caterina.funghi@live.com

## ABSTRACT

**Background**. In arid environments, plant primary productivity is generally low and highly variable both spatially and temporally. Resources are not evenly distributed in space and time (e.g., soil nutrients, water), and depend on global (El Niño/ Southern Oscillation) and local climate parameters. The launch of the Sentinel2-satellite, part of the European Copernicus program, has led to the provision of freely available data with a high spatial resolution (10 m per pixel). Here, we aimed to test whether Sentinel2-imagery can be used to quantify the spatial variability of a minor tussock grass (*Enneapogon* spp.) in an Australian arid area and whether we can identify different vegetation cover (e.g., grass from shrubs) along different temporal scenarios. Although short-lasting, the *Enneapogon* grassland has been identified as a key primary food source to animals in the arid environment. If we are able to identify and monitor the productivity of this species remotely, it will provide an important new tool for examining food resource dynamics and subsequent animal responses to them in arid habitat.

**Methods**. We combined field vegetation surveys and Sentinel2-imagery to test if satellite spectral data can predict the spatial variability of *Enneapogon* over time, through GLMMs. Additionally, a cluster analysis ('gower' distance, 'complete' method), based on *Enneapogon* seed-productivity, and total vegetation cover in October 2016, identified three clusters: bare ground, grass dominated and shrub dominated. We compared the vegetation indices between these different clusters from October 2016 to January 2017.

**Results**. We found that $MSAVI_2$ and NDVI correlated with the proportion of *Enneapogon* with seeds across the landscape and this relationship changed over time. Both vegetation indices ($MSAVI_2$ and NDVI) were higher in patches with high seed-productivity of *Enneapogon* than in bare soil, but only in October, a climatically-favorable period during which this dominant grass reached peak seed-productivity.

**Discussion**. $MSAVI_2$ and NDVI provided reliable estimates of the heterogeneity of vegetation type across the landscape only when measured in the Austral spring. This means that grass cover is related to seed-productivity and it is possible to remotely and reliably predict food resource availability in arid habitat, but only in certain conditions. The lack of significant differences between clusters in the summer was likely driven by

the short-lasting nature of the vegetation in the study and the sparseness of the grass-dominated vegetation, in contrast to the shrub vegetation cluster that was particularly well measured by the NDVI.

**Conclusions**. Overall, our study highlights the potential for Sentinel2-imagery to estimate and monitor the change in grass seed availability remotely in arid environments. However, heterogeneity in grassland cover is not as reliably measured as other types of vegetation and may only be well detected during periods of peak productivity (e.g., October 2016).

# INTRODUCTION

In Australian arid environments, plant primary productivity is generally low and highly variable both spatially and temporally, as a result of low levels of average rainfall that are aseasonal, and with high variance in the scale and timing of rainfall events (*Noy-Meier, 1973*; *Morton et al., 2011*). Even at a very local scale both annual rainfall, and the individual rainfall events can be very variable (*Acworth et al., 2016*). As well as presenting an ecological challenge for arid adapted organisms (*Letnic & Dickman, 2006*; *Morton et al., 2011*), this pattern of rainfall presents a significant challenge to research on Australian arid zone ecology. For example, in their analysis of 44 years of rainfall data collected over 17 rain gauges across around 380 km$^2$ of the Fowlers Gap Arid Zone Research Station, *Acworth et al. (2016)* demonstrated that the rainfall measured in one part of the station is often poorly correlated with that measured in other parts. In a number of representative years, some parts of the research station received more than twice as much rainfall as others, despite being less than 20 km apart (*Acworth et al., 2016*). Although at a wide scale a large rainfall event tends to have comparable effects across large areas, ecologically, the difference between these levels of rainfall across local patches will be profound. However, studies that explore the response of animals to rain typically use rainfall records that were taken at distance from the studied population (*Zann & Straw, 1984*; *Zann et al., 1995*), and in many cases use interpolated values between two weather stations that may be hundreds of kilometers from the study area (e.g., *Crino et al., 2017*; *Pavey & Nano, 2013*). To further confound the problem, the primary productivity on the ground is the result of complex interactions between soils, the water responses of different plant species, climate, seasonality, herbivores' foraging activity, and the total amount and speed at which the rain falls (*Reynolds et al., 2004*; *Morton et al., 2011*; *Nano & Pavey, 2013*). For example, rainfall events that are timed with the optimal growth phenology of certain species, can elicit a large response on those, but not others. This will promote great landscape heterogeneity at a small spatial scale, and complicate the link between rainfall and net-productivity of components of the arid zone community (*Watson, Westoby & Holm, 1997*; *Fernández,*

*2007*), in turn making it difficult to draw generalizations about rainfall pulses and arid productivity (*Reynolds et al., 2004*).

To improve our understanding of animal responses to rainfall, we need to better quantify local primary productivity, rather than using rainfall (or interpolated rainfall) as a proxy. Great progress has been made in satellite remote sensing, increasing our ability to study the general response of vegetation to rainfall in remote and arid areas. Data archives have become freely available (e.g., Landsat by NASA active since 1972, *Wulder et al., 2016*) and new satellites have been launched, specifically for land cover change monitoring. For example, the Sentinel 2 satellite, part of the European Copernicus program, provides data with a high spatial resolution (10 m per pixel) that is freely available (running between 2016 and 2028; *Skidmore et al., 2015*). In temperate climates, where the relationship between abiotic (e.g., rainfall, temperature, soil nutrients) and vegetation productivity is more predictable, satellite remote sensing information has been widely integrated into wildlife research, such as the response to plant phenology of both herbivorous (e.g., mule deer *Odocoileus hemionus*, *Hurley et al., 2014*) and non-herbivorous birds and mammals (reviewed in *Pettorelli et al., 2011*).

Despite this progress, the use of satellite remote sensing to address questions in animal ecology in the arid and semi-arid environment remains particularly challenging. Ground cover is heterogeneous and often sparse with a low vegetation cover and a high soil reflectance (*Nagendra, 2001*; *Okin & Roberts, 2004*; *Ren, Zhou & Zhang, 2018*). However, specific vegetation properties (e.g., density, biomass productivity) can be summarized by calculating spectral vegetation indices, usually combinations of two or more wavelengths/bands (reviewed in: *Kalaitzidis, Heinzel & Zianis, 2010*; *Xue & Su, 2017*). Many studies of arid areas have tried the most suitable vegetation indices to reliably estimate the variation in vegetation type and density across the landscape (e.g., Asia, *Kang, Wang & Liu, 2018*; Africa, *Mapfumo et al., 2016*; North and Central America, *Théau & Weber, 2010*; Australia, *Chen, Scientific & Gillieson, 2014*), finding that a vegetation index based on higher spatial resolution imagery better represent the actual situation on the ground. However, the use of a single vegetation index to summarize such a complex environment is not a reliable tool (*Okin & Roberts, 2004*; *Hamada et al., 2019*). Given the sparseness and patchiness of vegetation in the landscape, modern and highly resolved Sentinel 2 imagery (compared to previously used Landsat data at 30 m resolution) may provide a better tool to predict the productivity of different components of the vegetation, such as grasses. Data with a higher resolution will be more suitable in reflecting the heterogeneity of complex vegetation patterns on the ground, and the signal may be less overwhelmed by more dominant, and perennial shrub species such as *Acacia* spp. This is important because whilst there may be a general 'greening response' to rainfall, it does not necessarily provide evidence of a resource base for a particular guild of animals. For example, the nutritional requirements of the seed-eating zebra finch (*Taeniopygia guttata*) have been well characterized, with the seeds of dominant grass species such as *Enneapogon* spp. making up a considerable portion of their diet (>80%) in the Austral spring, triggering reproduction (*Morton & Davies, 1983*). However, these grasses often occur amongst patches of chenopod shrubs that provide little or no nutrition to this granivorous species, and may have a greening
response to rainfall, perhaps at other times of the year, that does not support growth of *Enneapogon* spp. Therefore, remotely measured indices derived from a suboptimal spatial resolution for the landscape in study, such as the Landsat-derived Normalised Difference Vegetation Index (NDVI, *Tucker, 1979*), may reveal a strong response that may be driven by variation in chenopod shrubs, and not reflect a response by other components of the vegetation, such as seed grasses, that are of relevance to granivorous species.

To date, most of the work on Australian extended arid grassland areas have attempted to relate satellite remote sensing information to areas dominated by perennial, sclerophyllous and tall grass species (*Chen, Scientific & Gillieson, 2014*). While Australian arid primary productivity is generally considered to be based on perennial, summer grasses like *Triodia* and *Plechrachene* spp. (*Dickman et al., 2014*); during the Austral spring, when the climate is most amenable for reproduction in many groups of animals such as birds (*Duursma, Gallagher & Griffith, 2017*), and while the summer grasses are still quiescent, the *Enneapogon* spp. are the most dominant grasses representing the main food source for a variety of animals (*Buckley, 1982*; *Hoffmann, 2010*). The *Enneapogon* genus is composed by minor tussocks (around 30 cm height), with 15 short-perennial species present in Australia (i.e., biannual, *Foulkes et al., 2014*; *Kakudidi, Lazarides & Carnahan, 1988*). The seed-productivity of *Enneapogon* grassland is commonly represented by a matrix of different *Enneapogon* species (hereafter generally called '*Enneapogon*') that peaks in favourable springs, in response to winter rains, and before the perennieal summer grasses start to grow.

By correlating the actual vegetation cover with remotely acquired spectral vegetation indices, it will be possible to determine the extent to which a remotely acquired signal represents vegetative growth in a particular area, and reflects different components of the plant community (*Chen, Scientific & Gillieson, 2014*). The seed-productivity and ground cover of *Enneapogon* in the arid landscape is not reliable and depends on the complex interactions with rainfall and other abiotic factors (e.g., temperature, soil nutrients). If the higher spatial resolution of Sentinel 2 imagery is able to capture the seed-productivity of *Enneapogon* in the arid landscape, this will provide new opportunities to understand the ecological response of seed-eating animals that primarily use these grasses over time and space. Here we combined field vegetation surveys and Sentinel 2 images that temporally matched the field sampling to directly test: (1) whether satellite remote sensing data (i.e., spectral vegetation indices that measure vegetation structure and cover) can be used to predict the spatial variability of *Enneapogon* seed-productivity in a heterogeneous and arid environment and (2) whether the spectral vegetation indices might be used to identify different vegetation cover (e.g., grass from shrubs) along different temporal scenarios. This would allow studies to remotely and reliably monitor the actual food availability dynamics in arid habitat, providing new insight to the study of arid zone animal population dynamics. The study objectives were addressed through two main field approaches. The first approach aimed to directly quantify *Enneapogon* seed-productivity across the landscape over an area of approximately 12.5 km$^2$, during October, December 2016 and January 2017. We tested the reliability of soil-adjusted and non-adjusted vegetation indices calculated from Sentinel
2 images with 10 m spatial resolution, to predict the spatial variability of *Enneapogon* seed-productivity across four months. In other words, we tested whether the satellite-derived indices can be used to predict times with low grass cover, and those times when grass cover and productivity are at their peak. The duration of the study from October 2016 to January 2017 permitted the capture of vegetation changes associated with the changing season and conditions in this arid environment, from an exceptionally productive Austral Spring in October (Fig. S1) to a dry Austral Summer in December–January 2017. We expected a link between Sentinel 2-derived vegetation indices and *Enneapogon* grassland at least in October, when the dominant grass was at its peak of seed productivity and ground cover. The second approach aimed to characterize the general vegetation composition of the landscape in the field, to group areas dominated by different vegetation (i.e., shrub dominated, or *Enneapogon* dominated) through a cluster analysis, and to test the reliability of the indices over four months between areas with different vegetation types. In this way, we quantified the spatial heterogeneity of the landscape and tested whether it will be possible, in the future, to assess the food productivity (i.e., *Enneapogon* seeds at its productivity peak) remotely. Determining the extent to which Sentinel's higher spatial resolution is capable of reliably representing vegetation heterogeneity in arid areas, across the landscape and over time, particularly for different parts of the vegetation community such as grasses, will guide the potential application of this imagery to become an important tool in attempts to understand the relationship between primary productivity and the ecological responses of animals in such an ecologically unpredictable environment.

## MATERIAL AND METHODS

### Study area and field surveys

The study was focused on an area - Gap Hills - located in the north of the Fowlers Gap Arid Zone Research Station ($31°05'13.1''$ S, $141°42'17.4''$ E), New South Wales, Australia. Fowlers Gap is one of the few long-term study sites in the Australian arid zone, where the population dynamics of several animal species have been monitored over long periods of time. Therefore, it is a strategic location to test whether food resource availability (i.e., primary productivity) can be reliably assessed remotely, to start linking animal dynamics and food availability in arid environments in the future. In October 2016, 36 quadrats of 10 m $\times$ 10 m were established in an area within 2 km of an artificial water dam (Fig. 1A), an important resource for the animals in the study area. In each quadrat, we identified all plants at the lowest taxonomical level possible (i.e., species or genus). For every identified plant, we estimated the percentage of vegetation–cover and noted the dominant genus of the overstorey, the understorey and for grasses. The total vegetation cover was estimated by considering the highest vegetation cover between the overstorey, understorey and grasses.

To quantify the variability of seed-productivity in the *Enneapogon* spp. across the landscape, we performed a weekly 50 m transect along the NE-SW line from the NE corner of every quadrat, during three weeks, between the 28th October and the 16th November 2016. The average of the data gathered over these three weeks corresponded to the October
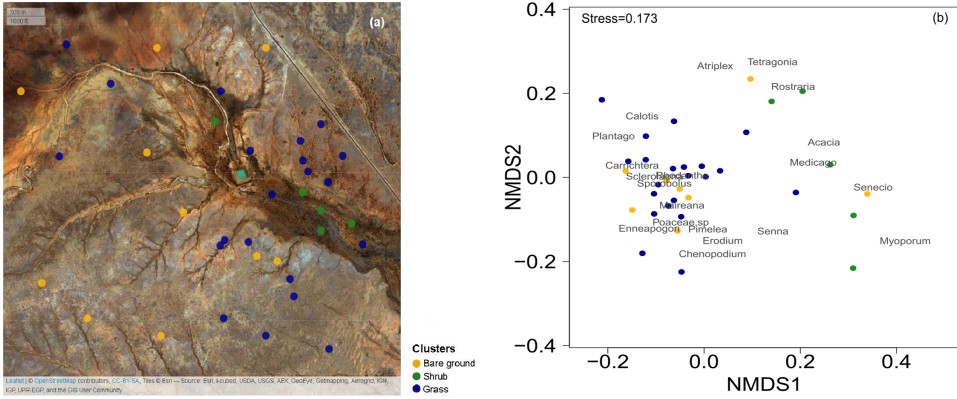

**Figure 1** **Map of the 36 surveys across the Gap Hills and the visual representation of the vegetation composition in the three vegetation clusters.** The spatial distribution of the 36 quadrat 10 m × 10 m surveys on ESRI satellite image, coloured according to the cluster analysis. Map credit: OpenStreetMap, 2016. Licensed under CC BY 3.0 SA. (b) Non-metric Multidimensional Scaling (NMDS) visual representation of the vegetation composition (genus of plants identified) and the quadrats surveyed in the three vegetation clusters considered. Grass (blue) and Shrub (green) vegetation categories were identified through a cluster analysis. The Bare ground group (yellow) was manually sorted being the total cover estimation less than 10% from the grass and shrub clusters. The stress value measures the goodness of fit of the data representation in multivariate space. Graphical distance represents similarity of the green and blue clusters. The location of the vegetation genus represents their co-occurrence.

values used in the statistical analysis. For every meter in each of our transects, we noted the name of the grass (genus level) and counted the number of spikelets with seeds and vegetative (without seeds) within 10 cm. Additionally, we randomly collected 50 spikelets with seeds of the most dominant grass genus, *Enneapogon*, from different individuals around the transect (and quadrat) area. In this way, the seed-productivity was calculated by estimating the proportion of *Enneapogon* spp. with seeds multiplying it with the average dry weight of one spikelet across the three weeks of sampling (g*seed/transect, analytical balance: Sartorius BP211D, Wood Dale, Illinois, 0.01 mg). In December 2016 and January 2017, we returned once to every location and performed the transect surveys. We estimated the seed-productivity of each transect location, that overlapped with each quadrat location, per each month. We multiplied the proportion of *Enneapogon* spp. with seeds by the average dry weight of one spikelet (from October–November spikelet collection).

In order to characterize the general vegetation composition of the landscape and identify areas dominated by different vegetation, we used the field-based data from our quadrats of October 2016 to estimate the presence of vegetation co-occurrence. We identified the most dominant 19 plant genera and built a matrix of the contribution of each to the plant cover across the plots. The cover-plot matrix was combined with the estimation of total cover and *Enneapogon* seed-productivity ('gower' distance in 'vegdist' for 'vegan' package and 'complete' method in 'NbClust' for 'NbClust' package in R; *R Core Team, 2014*, *Charrad et al., 2014*; *Oksanen et al., 2017*) to run a global non-metric multidimensional scaling analysis (NMDS) to identify different vegetation clusters (Fig. 1B): the *Enneapogon*-based ('grass') and the shrubs-based ('shrub', Table S1). Furthermore, an indicator species analysis was

run to statistically assess the strength of species-clusters associations. This analysis is based on permutation tests and an indicator value index that is the product between the relative frequency of occurrence for each species in each cluster and the relative abundance (cover) of each species in each cluster ('IndVal.g' methods in the 'multipatt' function in 'indicspeces' package, *De Cáceres & Legendre, 2009*). The quality of the overall ecological conditions is also determined by the amount of bare ground, and we manually built a third cluster selecting the 'bare ground' quadrats, defined by total vegetation cover estimation being less than 10% (Fig. 1). Later we used the same clusters to test the suitability of the satellite-derived vegetation indices to discriminate between a subset of quadrat locations identified inside these clusters.

## Satellite imagery

In order to best match the period of field data collection (*Enneapogon* seeds productivity sampled between 28th October and 16th November 2016) and the satellite-derived indices, we chose the imagery available from Sentinel 2 (Copernicus Earth Observation Program, Thales Alenia Space, ESA, 2015), with cloud free conditions for the study period, which was 29th October 2016. To analyze the temporal changes, from the Austral spring 'boom' (October–November) to the dry arid summer (December–February), the images selected were taken from the 18th December 2016 and the 27th January 2017 (both 0% cloud cover). We were unable to use images from November, because it was cloud affected. Images were projected into the WGS 84/UTM zone 54S coordinates reference system. For removing atmospheric effects, we applied the dark object subtraction procedure (*Chavez, 1988*) using the Semi-Automatic classification tool (*Congedo, 2016*) which is available as a plugin for QGIS (v 2.18.17, *QGIS Development Team, 2019*). For the high spatial resolution satellite images from all periods (October, December 2016 and January 2017), we calculated the two vegetation indices previously shown to perform well in arid and semi-arid areas (*Chen, Scientific & Gillieson, 2014*, Table S2). The Modified-Soil-Adjusted Vegetation Index (MSAVI$_2$, *Qi et al., 1994*) is adjusted for the reflectance of the exposed soil, whereas the Normalized Difference Vegetation Index (NDVI, *Tucker, 1979*) is not adjusted. Both of these indices use the near infrared (NIR, 0.84 $\mu$m, band 8) and red wavelength reflectance (Red, 0.66 $\mu$m, band 4, Table S2). The indices were calculated at native spatial resolution (10 m for MSAVI$_2$, NDVI), for each of the 36 GPS plots/transects points ($\pm$5 m, GPSMAP® 64s, Garmin, Olathe, USA), projected into WGS 84/UTM zone 54S coordinates reference system. Therefore, we obtained the value of MSAVI$_2$ and NDVI for a 10 $\times$ 10 m (proxies of pixel) square which included the GPS point used as reference for the field-based work.

## Statistical analysis

The spectral vegetation indices used are measurements of vegetation structure and cover, therefore, to validate the satellite remote sensing information against field based data across time, we considered the sampled transects that had a value of the *Enneapogon* seed-productivity larger than zero. Further, in order to balance the sample sizes of transects (larger in October), we additionally subset the transects considering the ones that in

October had higher than mean seed-productivity. Therefore, for these analyses the final sample size was 23 transects: three transects were repeated each of the three months in study, ten repeated twice and ten transects had no repeated measurements (i.e., $n_{observation}$ = 39). Our measures of *Enneapogon* seed-productivity (i.e., the product of the proportion of *Enneapogon* with seed and the average dry-weight of one spikelet) had a distribution not suitable to be modeled (neither as proportion nor Gaussian). For the analyses of the spectral vegetation indices validation, we used directly the proportion of *Enneapogon* with seeds over the total number of spikelets counted for each transect survey, which was highly correlated with the *Enneapogon* seed-productivity (rho=0.9, $P < 0.001$, $n = 39$). Having checked that the subset data met the assumptions for the spatial autocorrelation (Moran's *I* test) and the linear model assumptions, we built two separate Generalised Linear Mixed Models (GLMM) with proportional error distribution that is a binomial weighted for the total of the proportion (i.e., total number of spikelets counted for each transect survey). The proportion of *Enneapogon* with seeds of the subset locations was fitted as the dependent variable. Either $MSAVI_2$ or NDVI of the same quadrat (measured at 10 m × 10 m pixel as approximation of transect line surveys), the months (October, December and January) and their interaction were fitted as explanatory variables; transect location ID was fitted as random effect. Full models were always reduced by removing the least significant terms in a stepwise process, starting with the interactions as determined by likelihood ratio test between models (*Crawley, 2007*); until only significant terms remained and terms included in significant interactions. For GLMMs we used the package 'lme4' (*Bates et al., 2014*). Furthermore, we estimated a model quality index for each model, the conditional pseudo-$R^2$ (using 'r.squaredGLMM' function in 'MuMIn' package, *Barton, 2019*) that represents the variance explained by the entire model, including both fixed and random effects.

We tested the performance of the soil-adjusted $MSAVI_2$ and the NDVI in distinguishing between patches with different vegetation cover over time. We tested a best-case scenario in which we considered a subset of five quadrats that were most representative of the three different clusters (i.e., five of each type, Table 1) to balance the sample size and thereby excluding quadrats with mixed vegetation types that might otherwise confound these estimates. Having checked for significant differences between the quadrats representing each cluster (Table S3), a series of paired Wilcoxon's signed rank tests, separately for each month, was run between the subset of quadrats representing each cluster. Each of the three tests were Bonferroni adjusted for multiple comparison. For comparison tests we used the package 'stats' and all statistical analyses were conducted within the statistical environment of R (*R Core Team, 2014*).

## RESULTS

The spatial autocorrelation value, Moran's *I*, for the proportion of *Enneapogon* with seed ($n_{IDlocation}$ = 23) in study was −0.02, not significantly different from randomness (i.e., Moran's $I = 0$, $p = 0.8$). The proportion of *Enneapogon* with seed was related with both $MSAVI_2$ (Table 2, Fig. 2A, $n_{IDlocation}$ =23, $n_{observation}$ = 39) and NDVI (Table 2, Fig. 2B, n

**Table 1  Summary of number (n) of quadrats, *Enneapogon* seed-productivity (mean ± SD) and total vegetation cover (mean ± SD) of the clusters' subset used for the temporal analysis.**

|  | Bare | Grass | Shrub |
|---|---|---|---|
| n quadrats | 5 | 5 | 5 |
| *Enneapogon* seed-productivity (proportion*$g^{-1}$) | 0.03 ± 0.03 | 0.17 ± 0.03 | 0.001 ± 0.002 |
| Total vegetation cover (%) | 2.8 ± 1.1 | 26 ± 14.4 | 49.8 ± 26.8 |

**Table 2  Summary of the GLMMs.** Response variables, random terms, and variances (Var) are specified for each model. Values of fixed effects (estimated) and standard errors (S.E.) are *logit* estimates for the variables in the minimal adequate model. The $X^2$ (d.f.) and $P$ values represent the significance of the model, estimated from the comparison between the full model and the reduced one (without the interaction between fixed terms). Statistically significant $P$ values are marked in bold. Each model is based on a total of 39 observations of 23 transect locations.

|  | Response variable | Random term | Var | Fixed effect | Estimate | S.E. | $X^2$(d.f.) | $P$ |
|---|---|---|---|---|---|---|---|---|
| $MSAVI_2$ | Prop. *Enneapogon* with seeds | $ID_{location}$ | 3.91 | $MSAVI_2$: December (Intercept) | 2.66 | 1.04 |  |  |
|  |  |  |  | $MSAVI_2$ | −65.32 | 25.37 |  |  |
|  |  |  |  | January | 2.14 | 1.14 |  |  |
|  |  |  |  | October | −0.67 | 1.58 |  |  |
|  |  |  |  | $MSAVI_2$:January | −69.15 | 25.45 |  |  |
|  |  |  |  | $MSAVI_2$:October | 78.28 | 21.89 |  |  |
|  |  |  |  | $MSAVI_2$:Month |  |  | 43.65(2) | **<0.001** |
| NDVI | Prop. *Enneapogon* with seeds | $ID_{location}$ | 4.36 | NDVI: December (Intercept) | 2.75 | 1.02 |  |  |
|  |  |  |  | NDVI | −30.76 | 11.22 |  |  |
|  |  |  |  | January | 2.96 | 1.24 |  |  |
|  |  |  |  | October | −0.65 | 1.57 |  |  |
|  |  |  |  | NDVI:January | −47.55 | 14.2 |  |  |
|  |  |  |  | NDVI:October | 37.32 | 11.32 |  |  |
|  |  |  |  | NDVI:Month |  |  | 41.58(2) | **<0.001** |

$ID_{location}$ = 23, $n_{observation}$ = 39) but the relationship changed over time, as the interaction between vegetation index and month was significant for both models (Table S4). In October 2016, ($n_{IDlocation}$ = 21), the higher the proportion of *Enneapogon* with seed (correlated with seed productivity), the higher were the vegetation indices. The indices reflected the variability in *Enneapogon* with seed across the landscape (Table 2). In December 2016 ($n_{IDlocation}$ = 12) and January 2017 ($n_{IDlocation}$ = 6) the same relationship was negative, although the reduced sample size does not allow general interpretations (Fig. 2). The conditional pseudo-$R^2$ (model quality index) was 0.68 for the model with the $MSAVI_2$ and 0.7 for the one with NDVI.

The three clusters identified differed in their vegetation composition (Fig. 1, Table S1). The bare ground plots ($n = 10$), represented mainly bare soil with a total cover of less than 10% (3.5 ± 1.5% mean ± SD); the grass cluster was composed of plots ($n = 21$) in which *Enneapogon* and succulent smaller bushes (e.g., *Scleroleana* spp.) were indicator species (Table S1), while only few forbs and *Acacias* spp. appeared (Table S1). For the shrub plots ($n = 5$), located in close proximity to the water channel, the indicator species were herbaceous and leafy shrubs vegetation (e.g., *Acacias* spp., *Medicago* spp., Table S1).

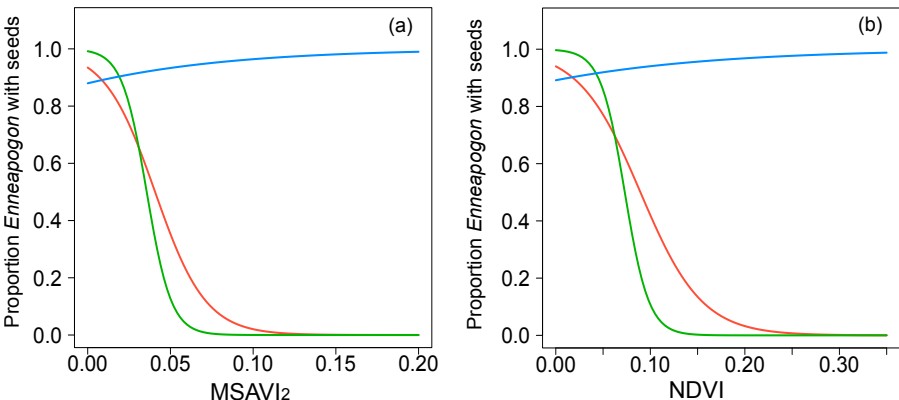

**Figure 2** **Graphical representation of the relationships modeled with GLMMs.** Binomial-GLMMs of the proportion of *Enneapogon* with seeds and the spectral indices $MSAVI_2$ (A) and NDVI (B) as predictor. Lines represent the predicted relationships for October 2016 (blue), December 2016 (red) and January 2017 (green).

**Table 3** **Summary of the paired Wilcoxon's signed rank between cluster subsets ($n = 10$ for each comparison).** Vegetation index (VI), cluster pairs tested (Pairs), month, coefficient test (Z) and significance (P) are specified for each comparison. Statistically significant values are marked in bold.

| VI | Pairs | Month | Z | P |
|---|---|---|---|---|
| | Bare-Grass | October | 0.6 | **0.03** |
| | Bare-Shrub | October | 0.7 | **0.01** |
| | Shrub-Grass | October | 0.7 | **0.01** |
| | Bare-Grass | December | 0.4 | 0.1 |
| **MSAVI₂** | Bare-Shrub | December | 0.6 | **0.02** |
| | Shrub-Grass | December | 0.4 | 0.2 |
| | Bare-Grass | January | 0.4 | 0.2 |
| | Bare-Shrub | January | 0.6 | **0.02** |
| | Shrub-Grass | January | 0.5 | 0.06 |
| | Bare-Grass | October | 0.5 | 0.06 |
| | Bare-Shrub | October | 0.7 | **0.01** |
| | Shrub-Grass | October | 0.7 | **0.01** |
| | Bare-Grass | December | 0.6 | **0.03** |
| **NDVI** | Bare-Shrub | December | 0.7 | **0.01** |
| | Shrub-Grass | December | 0.6 | **0.02** |
| | Bare-Grass | January | 0.4 | 0.2 |
| | Bare-Shrub | January | 0.7 | **0.01** |
| | Shrub-Grass | January | 0.7 | **0.01** |

With respect to NDVI, there were significant differences between bare ground and shrub vegetation patches, and shrub and grass patches in October, December and in January (Table 3). A significant difference between the NDVI of bare ground and grass patches was only detected in December (Table 3, Figs. 3D–3F), although the test value $Z$-score was just over the significance threshold.

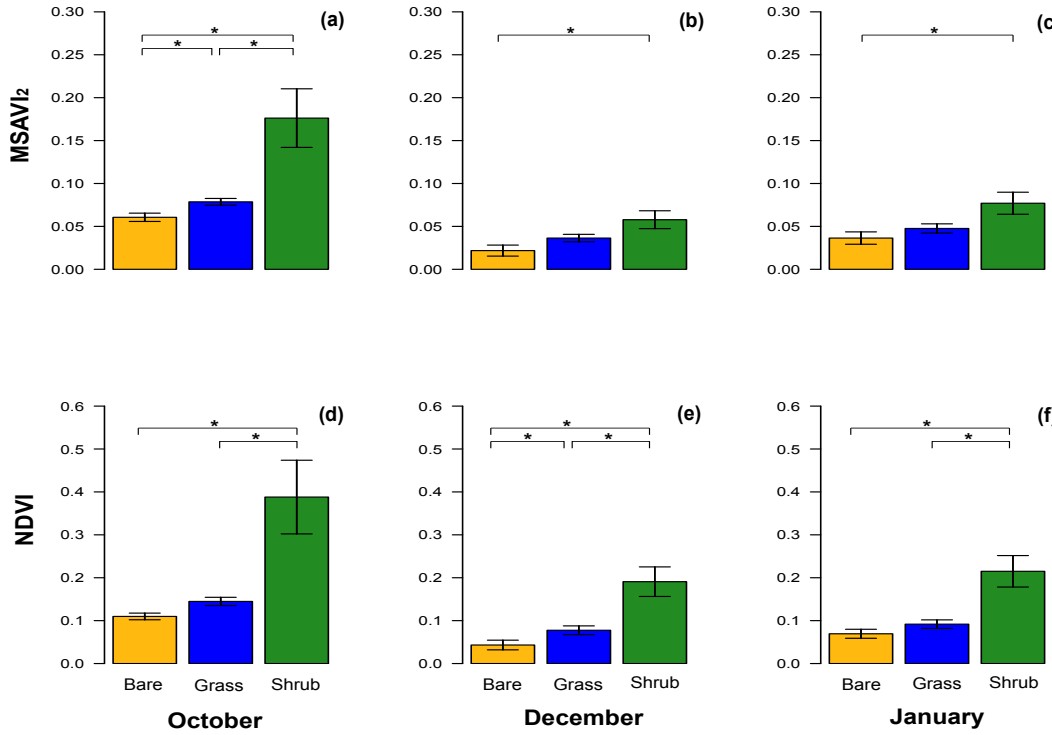

**Figure 3 Comparisons of spectral vegetation indices between clusters at different time.** Bar chart with mean ± SE to show the comparison of MSAVI$_2$ (A–C) and NDVI (D–F) extracted from the subset of the three clusters of quadrats with different vegetation type and density. The comparisons were performed for October 2016, December 2016 and January 2017. Significant differences are marked by * and the analyses performed were Wilcoxon's ranked tests and Bonferroni adjusted for multiple comparison.

The equivalent results for the MSAVI$_2$ index revealed significant differences across the three different vegetation cover in October, when the vegetation was at its peak (Table 3, Fig. 3A), but in December and January, during the dry Summer, the difference between bare ground and shrub was significant, whereas the comparisons between other clusters were not significantly different (Table 3, Figs. 3B, 3C).

## DISCUSSION

Using on-ground measures of vegetation structure and *Enneapogon* seed-productivity we tested the extent to which Sentinel 2 satellite images can be used to indicate habitat quality, and potential food resource availability (i.e., *Enneapogon*), in the spatially heterogeneous Australian desert and semi-desert. The results suggest it is possible to detect remotely differences in vegetation cover between patches over time, if these patches were previously chosen as dominated by specific vegetation types (e.g., grass, shrubs, bare ground), although with some limitations. We based the vegetation characterization focusing on a sample period (October 2016), where the whole vegetation in the study area was at the 'boom' phase of the characteristic 'boom and bust' cycle, as a consequence of good winter rainfall (*Morton et al., 2011*), and confirmed by this period having the highest NDVI value

since 2010 (Fig. S1). This allowed us to get a rare estimation of the vegetation cover and the *Enneapogon* seed-productivity across the landscape during extremely favorable conditions.

Our results showed that both spectral vegetation indices were positively related to the proportion of *Enneapogon* with seeds (highly correlated with seed-productivity) across the landscape, at least in October. Furthermore, the cluster analysis showed that *Enneapogon* was most likely to be spatially associated with succulent shrubs *Scleroleana* spp. On the other hand, areas closer to the semi-permanent dam were dominated by *Acacia* and a few species of forbs. The distribution of vegetation supported previous studies which found the presence of artificially dam water to influence the spatial distribution of vegetation types, perhaps through the effect of heightened grazing (around dams) that favors shrubland over grassland (*James, Landsberg & Morton, 1999*). In our field site the water dam is also along a dry-creek system that might further influence the vegetation response found, with the taller and greener vegetation distributed along the ephemeral creek. Actually, dry creeks have been demonstrated to increase the soil water content in the adjacent area, by prolonging the effect of flood after rain events, and triggering a strong vegetation response (*Kingsford, Curtin & Porter, 1999*; *Morton et al., 2011*). Therefore, it is likely that these environmental structures and their associated effects interact to shape the spatial vegetation variation detected by the Sentinel-derived indices, even at local scales.

The other aim of the present study was to validate the relationship between spectral vegetation indices and vegetation patch cover over time. Plots that equate to the size of a pixel in the satellite imagery are rarely comprised of pure vegetation cover of a single type of vegetation in arid habitat. However, to validate the potential of Sentinel 2 imagery in arid zone ecology, it is important to discriminate and monitor the cover of different vegetation patch types over time. Thus, our results related the grass cover with seed-productivity, allowing its monitoring from the space, knowing that the satellite-derived vegetation indices will detect only the peak of cover/productivity. This suggests that in the future it will be possible to monitor the seed-availability across Gap Hills remotely, if one can focus on patches that are known to contain grass, and ignore the 'greening' signal from areas of shrub cover. By doing this, and estimating the spectral vegetation indices from these potential areas of grass, the imagery should be able to discriminate effectively between times with low grass cover, and those times when grass cover and productivity is at its peak. The lack of temporal reliability shown in December and January could be due to the short-lasting nature of the vegetation, rather than an actual unreliability of the satellite sensitivity signal. The spectral vegetation indices we used are designed to detect greenness differences (i.e., wavelengths' spectra) emitted from the vegetation (*Kalaitzidis, Heinzel & Zianis, 2010*; *Xue & Su, 2017*). Therefore, they are expected to be influenced by vegetation phenology. Additionally, when we considered the clusters' subset that best characterized the different dominant vegetation kind (or bare soil), we found that only in October the $MSAVI_2$ was able to distinguish between areas with high *Enneapogon* seed-productivity, bare soil and dense shrub vegetation. However, the difference between the most stable clusters, the shrub and bare ground quadrats, held even when the environmental conditions became more arid (January 2017), despite the reduced power of the analysis at that time due to the reduced sample size of the quadrats' subset. Other satellite remote sensing techniques and

indices might be more suitable for detecting specific changes in phenology accounting for changes in water, chlorophyll content and plant litter (e.g., *Berry & Roderick, 2002*; *Szabó, Gácsi & Balázs, 2016*).

We detected a slight difference in performance between the soil-adjusted MSAVI$_2$ and soil-unadjusted NDVI along time and across clusters, with the NDVI always detecting the difference between shrub and bare ground and shrub and grass patches, while MSAVI$_2$ mainly distinguished between bare ground and shrub patches. Other studies in arid habitats, found that at low vegetation cover (<30%), the unadjusted vegetation index performed better than the adjusted one (e.g., *Ren & Feng, 2014*). Additionally, a previous study tested the use of several vegetation indices across the whole Fowlers Gap Research Station, using Landsat TM and intensive ground surveys (six transects of 3 km in 49 sites and 147 3 m × 3 m quadrats, (*Chen, Scientific & Gillieson, 2014*) found that both the MSAVI$_2$ and NDVI were reliable only in wet conditions. Our results showed that the MSAVI$_2$ andNDVI reliably discriminate between high vegetation cover and bare ground areas over time.

Although the overall size of our study area was much smaller than the area used in other studies in the literature, the analysis of Sentinel's imagery was able to capture the variability across the landscape, when the vegetation was at its peak. Sentinel's higher spatial resolution has been shown to provide higher accuracy in the retrieval of vegetation phenology of an heterogeneous landscape, like a Dutch barrier island, than medium-resolution sensors (e.g., MODIS, *Vrieling et al., 2018*). Sentinel-derived NDVI seems to better reflect soil moisture condition of areas in extreme drought conditions than lower resolution sensor-based NDVI (*West et al., 2018*). These results suggest that Sentinel's higher spatial resolution better represents ecologically heterogeneous environments such as the arid ones studied here. This is important, because to understand the responses of particular animals to their environmental fluctuations, we need to focus on the relevant patches of vegetation. Some animals, such as kangaroos and others large herbivorous mammals, may respond to the general greening response detected by the difference between shrub patches and bare ground, because that will reflect the vegetative status and abundance of shrubs –their food. However, for animals that are strictly granivorous i.e., which dependent solely on the productivity of patches of grass that are detectable only when at peak seed-productivity, as demonstrated here, a challenge remains.

An optimal spatial resolution allows a reliable estimation of the structural characterization of plant association (e.g., species identification), maintaining information on vegetation types and abundance, which relates with spectral heterogeneity (*Nagendra, 2001*). Thus, whilst in homogenous landscapes (e.g., woodland) a lower spatial resolution may be optimal, more complex and patchy environments (e.g., arid areas) may require a higher spatial resolution to be optimal. For example, to analyze the spatial movement of Topi antelopes (*Damaliscus lunatus*) and vegetation phenology, MODIS images (250 m spatial resolution) revealed a pattern that AVHRR (5 km spatial resolution) did not detect (*Bro-Jørgensen, Brown & Pettorelli, 2008*). Furthermore, understanding animals' individual variation in habitat use is a key step to reveal their variation in breeding phenology and to test the possible evolutionary responses to climate change (*Dall et al., 2012*; *Merilä & Hendry, 2014*). For example, a study over 12 years showed a match between the phenology

of oak trees across a woodland, with caterpillar availability, and the individual variation in breeding phenology of a population of great tits (*Parus major*) and blue tits (*Cyanistes caeruleus*, *Cole et al., 2015*). This means that it is possible to evaluate the synchrony between trophic levels at a scale relevant to the individuals in a population, identify the environmental cues used by animals to time their breeding and, consequently, to understand how selection act on these phenological traits (*Durant et al., 2007*; *Cole et al., 2015*). In this context, our results, showing the Sentinel-derived indices' discrimination of vegetation types, suggest that 10 m may be a sufficient resolution to catch some key components of landscape variation in arid environments and, therefore provide a tool for studying animal responses to environmental phenology at a scale relevant to individuals.

## CONCLUSION

Overall, we showed that the spectral vegetation indices $MSAVI_2$ and NDVI, calculated from the freely available and spatially highly resolved Sentinel 2 satellite images, are able to provide reliable estimates of both spatial and temporal vegetation cover, with some limitations. The temporal reliability seemed to reflect the short-lasting nature of the key plant species present. However, the fine discrimination between vegetation types (i.e., *Enneapogon* versus shrubs or bare ground) was accurate only during an ecologically productive period, when this small dominant species reached their peak. Less subtle vegetation type discrimination (i.e., bare ground vs shrub vegetation) held over time, despite spatial heterogeneity and prolonged dry conditions that can reduce the 'green-vegetation' signal, especially in an arid environment. Therefore, by previously identifying patches covered by specific vegetation (e.g., performing vegetation surveys when the vegetation in study is at its peak), using Sentinel 2-derived spectral vegetation indices, it might be possible to track their responses to abiotic conditions (e.g., rainfall). In this way, the actual condition of food resource availability (i.e., grass) across a habitat might be estimated at high spatial resolution. Consequently, it would be possible to study how animals respond to this unpredictable environment (including local rainfall variability, *Acworth et al., 2016*) overcoming the complex relationship between rainfall and vegetation responses, which make generalised predictions quite unreliable (*Reynolds et al., 2004*). We suggest that the data generated by the Sentinel 2 will provide a reliable estimation of habitat condition (and food availability) over time, with some limitation with respect to some components of the vegetation. Satellite remote sensing of vegetation will enable better long-term studies of animal responses to the unpredictable conditions in the arid zone.

## ACKNOWLEDGEMENTS

We thank two anonymous reviewers for improving this manuscript with clear and supportive observations; Karen Marais, curator of the Downing Herbarium at Macquarie University, for help and advice with the field survey protocol and for assistance with plant identification; Professor Mark Westoby for the helpful advice about grass productivity; the Ecology Lab at Macquarie University for all the field work and biomass measuring

materials; the director and the manager of Fowlers Gap Research Station for their support; the Dowling family for their kind support in harsh conditions.

### Funding

This work was supported by the 'Deutsche Forschungsgemeinschaft' (SCHU 2927/3-1 to Wiebke Schuett and Simon C. Griffith), an ARC Future Fellowship Grant to Simon C. Griffith (FT130101253) and the Joint Degree International Macquarie University Research Excellence Scholarship to Caterina Funghi (iMQRES - 2016204). The funders had no role in study design, data collection and analysis, decision to publish, or preparation of the manuscript.

### Grant Disclosures

The following grant information was disclosed by the authors:
Deutsche Forschungsgemeinschaft: SCHU 2927/3-1.
RC Future Fellowship Grant: FT130101253.
Joint Degree International Macquarie University Research Excellence Scholarship: iMQRES - 2016204.

### Competing Interests

Simon C. Griffith is an Academic Editor for PeerJ.

### Author Contributions

- Caterina Funghi conceived and designed the experiments, performed the experiments, analyzed the data, prepared figures and/or tables, authored or reviewed drafts of the paper, and approved the final draft.
- René H.J. Heim and Jens Oldeland conceived and designed the experiments, analyzed the data, authored or reviewed drafts of the paper, and approved the final draft.
- Wiebke Schuett and Simon C. Griffith conceived and designed the experiments, authored or reviewed drafts of the paper, and approved the final draft.

### Data Availability

The raw data is available in the Supplementary Files.

### Supplemental Information

Supplemental information for this article can be found online at http://dx.doi.org/10.7717/peerj.9209#supplemental-information.

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
