# Peer review of "Estimating food resource availability in arid environments with Sentinel 2 satellite imagery"

_PeerJ, doi:10.7717/peerj.9209_

## Round 0.1 · original submission · Major Revisions

After careful consideration, I feel that it has merit but does not fully meet PeerJ publication criteria as it currently stands. Therefore, I invite you to submit a revised version of the manuscript that addresses the points raised during the review process.

Please provide either line numbers, the revised sentences themselves, or, preferably, both, in the response to reviewers document.
Please do grammar/vocab checking throughout the text.

Please provide a sound link between MSAVI and seed productivity.
Fig. 1: labels are too small and the graphics must be improved.

Abstract could be improved.

Description of Braun-Blanquet index needs to be expanded.

Reviewer 1 ·

Basic reporting

Major comments (in order of importance)

1) At many points, the sentence structure is very unusual (e.g. lines 180-182, 262, 275, 328), making it difficult to understand the content. Additionally, there are a considerable number of grammar/vocabulary errors (e.g. lines 55, 64, 107, 122, 190, 216) and punctuation errors (e.g. lines 65, 66, 162, 315) throughout the text. I have provided several examples for each of these issues, but since they occur frequently throughout the text, I am unable to highlight every single instance. I think the manuscript would greatly benefit from extensive editing, especially by any authors (or colleagues) that are native English speakers.
2) The title is evocative and easy to understand but, in my opinion, does not fairly represent the work that is carried out in this paper. Specifically, I don’t think the work presented here integrates remote sensing with behavioural ecology – this implies that there is some integration of movement or other animal behaviour related data, which does not occur here. The work does, however, make an interesting contribution to the remote sensing of habitat qualities of particular species (namely, food resources), which I think ought to be reflected in the title.

Minor comments (in no particular order)

3) I disagree with the use of “remote sensing” to refer to the satellite data-derived vegetation indices throughout the text, since “remote sensing” encompasses many more modes of data acquisition (e.g. camera traps, acoustic sensors, telemetry) that are neither used nor discussed in this study. I suggest this is replaced with “satellite remote sensing” instead.
4) The authors use the term “barren” throughout the text to refer to bare ground. However, barren implies the that a patch of land is sterile (e.g. lacking a seed bank), but this cannot be directly deduced from low vegetation cover. I suggest the authors use “bare ground” instead.

Experimental design

Overall comments: The Methods section does not provide enough evidence that the statistical analyses were appropriate. Additionally, clarification is needed in some places to allow replication.

Issues which require re-analysis (in order of importance):

1) In the Introduction, the authors state that their first aim is to determine whether “remote sensing data […] can be used to predict the spatial variability of Enneapogon seed-productivity [sic]” (line 132). However, the authors use Enneapogon seed productivity as an explanatory, not a dependent, variable in the models. This means that the goodness of fit measure (RMSE) describes how well spatial variability in Enneapogon seed productivity explains variability in the remotely sensed index, and not vice versa. It is not clear why the authors chose this approach. I suggest that the authors rerun their models using Enneapogon seed productivity as the dependent variable.
2) The same rationale applies to the analysis of how well vegetation indices discriminate between the three main land cover classes (bare ground, grass-dominated and tree-dominated). Again, satellite-derived vegetation indices are treated as the dependent variable, whereas the study is interested in understanding “how Sentinel 2 information can be used to estimate habitat condition in a spatially heterogenous environment […]” (line 260-261), implying that their usefulness as an explanatory variable is of interest instead.
3) The authors state that Generalised Least Squares (GLS) models were used to account for the “non-linearity of the natural spatial distribution of the vegetation” (line 208). It is not clear what is meant by this, given that GLS approaches do not relax the assumption for linear relationships between predictor and dependent variable (they do however relax the assumption that there is no correlation between residuals). Since the authors provided their raw data in a very accessible and very easy to understand format (thank you!), I was able to fit the GLS models myself. The diagnostic plots I obtained showed a worrying increase in the spread of standardized residuals for large fitted values for models using seed productivity as explanatory variables. I am not convinced, therefore, that the GLS models are currently an appropriate way to characterise the relationship between the satellite-derived vegetation indices and the vegetation parameters obtained in the field. I suggest the authors explore alternative models that fit the data better (e.g. Generalised Linear Models or non-parametric tests).
4) It is not clear why the analysis of how much the satellite-derived vegetation indices differ between the three land cover types is limited to the five most representative plots for each type. If the question is to which extent vegetation indices can discriminate between different land cover types, this is likely to bias the results towards suggesting more successful discrimination. I propose that the authors rerun the analyses using all available vegetation plots.
5) Currently, 18 separate Wilcoxon tests are run to identify which land cover types have significantly different vegetation index values, but no penalisation for multiple testing seems to have taken place. This could be rectified by using e.g. a Bonferroni correction, or consider a mixed ordinal regression model with a vegetation index as explanatory variable, and month as a random effect, to reduce the amount of statistical tests carried out.

Clarification needed (in no particular order):

6) Please elaborate on your choice of study site, specifically why the vegetation quadrats were established close to an artificial water dam (see line 152).
7) The description of how the Braun-Blanquet index was adapted is difficult to understand (lines 155-156). Specifically, could you clarify what criteria you used to determine the new threshold values?
8) Please clarify whether the machine described in lines 169-170 is a scale.
9) Please elaborate on the difference between atmospheric correction and removing absorption effects (line 193). Which algorithm was used for the latter?
10) Were the GPS points of the quadrats (line 203) taken in the middle of the quadrants? Please explain how you ascertained that each 10m by 10m quadrant only overlapped a single Sentinel 2 pixel, given that the spatial error of the GPS was 5 m.
11) The authors find significant Moran’s I values for both satellite-derived vegetation indices; however, the values were quite low, and though significant, are unlikely to be important. Given that the best models for each vegetation index were those without a spatial correlation factor, please add an explanation for why the inclusion of the spatial correlation in the final model was justified.
12) The description of how the authors identified the 5 most representative plots for each land cover class (bare ground, grass-dominated, tree-dominated) is unclear (lines 228-231). I assume the five plots with the highest relative coverage of bare ground, grass and Acacia (respectively) were used, but I am not quite sure after reading the description.

Validity of the findings

The analyses and findings as presented in the Results and Discussion section do not clearly correspond to the stated aims of this study. I have four major comments, in no particular order of importance:

1) The authors state in the Introduction that the present study has two aims: 1) find out to which extent Enneapogon seed productivity can be predicted by remotely sensed vegetation indices and 2) determining to which extent this relationship is robust over time. However, they then introduce apparently unrelated analyses in their Methods (the relationship between remotely sensed indices and total vegetation cover, the difference in vegetation indices between three broad land cover types). I suggest the authors either omit these analyses (and the discussion thereof), link these analyses more explicitly to their hypotheses, or revise the stated aims of this study.
2) Additionally, they do not explicitly test to which extent the seed productivity-vegetation indices relationship holds over time. The Introduction should be amended accordingly.
3) It is not clear, from the Discussion, to which extent the validity/generality of the results is affected by the fact that the year in which the study was carried out (2016) had exceptionally high NDVI values (line 267), given that the authors conclude that peak “greenness” was important in distinguishing between land cover types (lines 360-362). The discussion should be expanded accordingly.
4) The conclusion states that “data generated by Sentinel 2 will provide a reliable estimation of habitat condition […] over time, with some limitation with respect to some components of the vegetation.” (cursive added for emphasis). There is however no discussion of what constitutes the limitations or the vegetation components in question; adding this would enhance the discussion considerably.

Additional comments

What interested me most about this paper was how field- and satellite-derived data were combined with an in-depth understanding of context-specific ecosystem processes to address a question of global interest – how can we measure habitat quality from space? Though the manuscript requires substantive revision (as detailed in my comments above), I believe a revised version has the potential to make an interesting contribution to our efforts to link satellite-derived data to estimates of habitat quality.

Reviewer 2 ·

Basic reporting

The paper: “Integrating remote sensing and behavioral ecology in the arid zone is basically using a clear and unambiguous English language throughout the text”, though I found a few sentences difficult to read, which I marked in the specific comments. The introduction and background is well written and the paper structure conforms to the peerJ standards. I suggest to alter the title and to avoid behavioral ecology in the title. It might be misleading since no behavioral data set is included into the analysis, even though the connections are argued in the introduction and discussion. The figures might (I am unsure if a few comments are due to the pdf version) need some improvement which you can find in the specific comments. The submitted raw data to my opinion needs to be enhanced by the vegetation data.

Experimental design

The research topic is well within the scope of the journal and the statistical analysis is generally sound. Though the cluster analysis, as well as the NMDS are not well explained in the methods section.

Validity of the findings

Throughout the manuscript I did not see the functional link between MSAVI and seed productivity. Could you please provide a sound mechanistical link of why MSAVI should be able to predict seed productivity? Here I am also missing the results for the NDVI to compare with. You chose a linear model but actually from Tab.2 you can see that seed productivity is quite low in the green cluster. You determined the plants only on genus level, can you argue why?

Additional comments

Table 2: Check for significant differences.
Table 3: From which month is the NDVI, indicate level of significance in the caption.
Table S1: Please modify the caption, what is the meaning of numbers (indicator value, relative abundance or relative frequency)? Order the table and draw a box around each cluster. Here you specify only two clusters but in the rest of the analysis you have three clusters (are there no plant species in barren?). Here you use the word vegetation type to specify the life form, in the rest of the paper you probably mean vegetation composition types…. if so be more specific.
Fig. 1: This figure is difficult to read at 100% zoom, labels are too small and graphics are not eqally sized. Please specify the meaning of the point sizes in the map. Use crosses to indicate the location of genera in the ordination diagram. Check if you always use a space between number and unit 10 x 10 m. ESRI instead of ENSRI. Clusters are not always named coherently ( Acacia/trees /green)? You could also think of more meaningful names referring to vegetation types. I also suggest to add a small, concise description of the clusters to your results, since you argue that you researched vegetation composition.
Fig.2: R² and or RMSE values are missing in the figure, coordinates are not readable, no scale bar, and the legend at my computer appears rather blackish. Do not use your model abbreviations as labels “Enn. Seed productivitity” and add proper units to the axis label.
Abstract
Background: the last sentence is incomplete or needs rephrasing.
Methods: I suggest to move the research questions to the Background and add information about your design, GLS etc. in methods. More specific information
Results: change vegetation density to vegetation cover. Be more specific, add model quality parameters to prove your statement.
Discussion and Conclusion are missing in the abstract.
Text:
L35you did not specifically test for discrimination…change the wording.
L39: You did not include any animal ecological data and within the abstract the connection between behavioral studies and remotely sensed data remains unclear, so please rephrase L22-23 (include more specific examples or skip it) (like: remote sensing data offers the possibility to model important predictors for animal behavior).
L54: 20 km
L84: rephrase clumped and dispersed at the time sounds funny
L94: spatial resolution-derived, sounds unclear to me
L115: be consistent either latin names or trivial names
L 188-190: Is the December belonging to the boom phase? Or did you select 2 images from arid summer to test model transferability? Please be more precise.
L194: Please briefly state, which technique you used for removing the absorption features to make your workflow also understandable to e.g. ARC GIS or R users.
L196: Sentence remains unclear: specify what you mean by high spatial resolution wavelength.
L213: You calculated the direction and amount of autocorrelation but for which model did you use it as dependent variables and why? Please add one more sentence to clarify.
L220: AIC, is it spelled out before?
L222: error or deviation? (RMSE /RMSD?)
L225: I suggest to rephrase since it does not match your research questions, since you never explicitly modelled vegetation composition, just Enneapogon abundance and seed productivity.
L226-233: Here, you could also apply the concept of distance (Jeffries-Matusita) that would fit more to the concept of discrimination.
L243: I am not familiar with GLS but I expected to see some sort of quality indicator for the relationship (R²), or the RMSE printed inside the figure.
L248: I suggest to use a more careful wording e.g. rephrase: the NDVI is significantly different between the three vegetation types (instead of distinguishes)
L250: rise?
L261: I suggest to use a more careful wording: instead of simply writing habitat condition I suggest to first of all state the seed productivity which you directly assessed and then argue why this implies habitat condition.
L313. High vegetation cover
L319: Could the capacity to discriminate between the classes be due to the selection of the five most barren / covered plots?
L328: Delete: that can browse on shrubs
L263: Did you really compare the MSAVI in October to the MSAVI in December and January? Please be a more specific in the wording.
L270: I suggest to stick to coverage, however I think you have a rather inhomogeneous vegetation pattern in your data with sharp contrasts between barren and green patches. Please be also more careful you did not directly assess the quality of your field estimations, so I would not argue that simply because my model is good my vegetation coverage estimates were good.
L274: So your data shows that Enneapogon is most abundant in the grass cluster. This sentence needs a little rephrasing to be sure in which cluster Enneapogon is most dominant.
357: satellite imagery
358: I propose: vegetation cover instead of density

---

## Round 0.2 · Minor Revisions

Thanks for the revised MS. The topic of the proposed manuscript is interesting. The reviewers and I have found your revision quite satisfactory, however, some minor revisions to your manuscript is required. Therefore, I invite you to respond to the reviewer's comments and revise your manuscript accordingly.

One more time, please provide either line numbers, the revised sentences themselves, or, preferably, both, in the response to reviewers document.

Reviewer 1 ·

Basic reporting

All line numbers refer to “Manuscript with tracked changes”.

Major comments (in order of importance)

It is not clear to me, from the Introduction, how the vegetation cluster analysis contributes to answering the central question of the paper (namely, whether satellite-derived indices can be used to predict food resource availability in arid environments). I suggest either adding an explanation to the Introduction (e.g. before l. 162) or omitting this part of the analysis from the paper.

There are still irregularities and mistakes in grammar and vocabulary that sometimes make it difficult to understand the paper. There are too many instances to list all of them, but for examples, see e.g. l. 60, 130, 185-186, 198, and the caption of Figure 1. As a result, I would encourage a second round of thorough editing.

Minor comments

Figures 1 are 2 quite blurry, so I suggest making a more highly resolved version. Additionally, the points in Figure 1b would be easier to see if they were a bit larger. In Figure 3, I think using a lighter colour would be better for the middle bar, as it is difficult to see the lower standard error limit.

It is not immediately obvious to the reader that the top half of Table 2 refers to one satellite-derived index, and the bottom to another, so it would aid understanding if that was somehow made more obvious. I am also a bit confused as to the meaning of the chisquare test – could you add some explanation in the Table caption?

l. 41-42: It is not clear 1) which vegetation types differ from each other, and 2) in what quality they differ -- could this be clarified?
l. 45: Does this refer to “heterogeneity in seed production depending on vegetation type”? If not, how is vegetation heterogeneity relevant for predicting food resource availability?
l. 47: I am not sure to what “temporal unreliability” refers. There is also no mention of how useful satellite data-derived indices are for predicting food resource availability. Since this is a key question of the study at hand, I would suggest adding this to the Abstract.
l. 145: What does primary seed productivity refer to? Does this mean the grasses set seed once at the beginning of a given season, and then once again afterwards?
l. 147: I think this needs a little more detail about the hypotheses/predictions (I do not understand why vegetation type matters in this context for instance, see Major Comments). Essentially, I think this needs to answer the question: why is it relevant whether or not satellite-derived indices can discriminate between different vegetation types to predict food resource dynamics?
l. 176-179: Since the paper does not compare the relative utility of satellite remote sensing and rainfall data for predicting resource availability, or integrate said animal population dynamics, I am not sure why this motivated the choice of study area. Perhaps there are ecological characteristics that make this area interesting?
l. 182: It still is not clear to me why the transects were established close to the dam.
l. 188: Does this mean that each transect was sampled once per week, over a period of three weeks, in October? And then once again in December and January respectively? I think this needs to be clarified in the text.
l. 213: All vegetation types are used in the analysis, rather than a subset – I think the subsetting refers to the within-cluster replicates?
l. 246: I am not sure to what “October relative” refers – this is a problem when replicating the analyses because it is not possible to identify which observations were included in the analysis.
l. 252: I am not sure what “grass count” means here.
l. 253: The quadrats are 10 m by 10 m (i.e. refer to a single Sentinel 2 pixel), whereas the transects used to estimate seed productivity are 50 m long – I think this needs an explanation of why average NDVI and MSAVI2 values across the entire transect were not used instead.
l. 300: Repetition of Sentinel 2.
l. 303: I am not sure what aspect “variation” refers to.
l. 338: The word “satellite” is missing.

Experimental design

All line numbers refer to “Manuscript with tracked changes”

Bonferroni corrections have not been applied to Table 3, as far as I can see. These tests require a correction by 9 for each dependent variable (as there are 9 statistical tests per dependent variable), meaning the new alpha level is 0.0056; none of the pairwise comparisons are statistically significantly different after Bonferroni correction. The discussion needs to be amended accordingly (l. 319 -333).

Bonferroni corrections not applied to Table S3. This requires a correction by 3 (3 tests each per dependent variable), meaning the new alpha level is 0.017; none of the pairwise comparisons are statistically significantly different after Bonferroni correction.

Validity of the findings

All line numbers refer to “Manuscript with tracked changes”

Major comments, in no particular order

The results reported from the GLMM do not correspond to the Discussion – specifically, the effect sizes for both satellite-derived indices are negative (see Table 2) but are described as positive in the Discussion (l. 311) and appear positive in Figure 2. I suggest checking that the correct effect sizes have been reported, otherwise the Discussion has to be amended.

The effect direction of both satellite-derived indices does not correspond to Figure 2. For instance, the intercept is negative for both models but eyeballing Figure 2 indicates that intercept is positive. Could you double check that the correct effect sizes are reported in Figure 2?

The effect size of both satellite-derived indices is smaller than the standard error for these terms (see Table 2), implying that the link between seed productivity and these indices is highly variable and rather weak. In my opinion, the findings are not accurately represented unless these limitations are addressed, either in the analysis or the discussion.

I do not agree with the conclusion that the satellite-derived indices presented here “provide reliable estimates of both spatial and temporal vegetation cover“ (ll. 393), since only pure vegetation types were selected to test whether satellite-derived indices can distinguish between them (l. 263). However, if pure pixels are relatively uncommon across the landscape, this would imply that the satellite-derived indices are not useful for predicting vegetation type for a lot of the area. I suggest adding an explanation for why pure pixels are expected to be common across the landscape.

From the Discussion, it is not clear why and how distinguishing between vegetation types helps predict food resources (see also comments in Basic Reporting). The paper is primarily focused on grass seed productivity, and it is not obvious how being able to discriminate between shrub and bare ground helps map this food resource. I would add an explanation of this to the Discussion

Minor comments

l. 316-325: The discussion of what drives shrub distribution across the landscape is unrelated to the research questions set out in the Introduction. I’d consider deleting or explaining relevance to research question in more detail.
l. 326: This research aim was not mentioned in the Introduction. I suggest adding this accordingly.

Additional comments

Thank you for new revised version! There are, as you have seen, still several areas where I think significant change needs to happen for this paper to effectively communicate its message. However, I believe it will be very straightforward to carry these out, and I am looking forward to seeing the revised version.

Reviewer 2 ·

Basic reporting

The manuscript clearly benefited from the review process. However, I think the language is still complicated or grammatically wrong in some cases: L 326; Caption Fig. 3, L387; as well as there are some minor spelling mistakes, such as missing spaces: Caption Figure 1.; L45,

Moreover, it was difficult to catch the narrative of the paper easily: the research questions are framed very differently:
e. g.: Abstract:
Here, we aimed to test whether Sentinel 2 imagery were able to accurately assess the spatial and temporal variability of a minor tussock grass (Enneapogon spp.) in an Australian arid area.

Introduction:
1) whether satellite remote sensing data (i.e. spectral vegetation indices) can be used to predict the spatial variability of Enneapogon seed-productivity and
(2) whether this prediction holds over time.

Discussion:
Using on-ground measures of vegetation structure and Enneapogon seed-productivity we tested
the extent to which Sentinel 2 satellite images from Sentinel 2 can be used to indicate habitat
quality, and potential food resource availability (i.e. Enneapogon), in the spatially heterogeneous
Australian desert and semi-desert.

So in my opinion some careful minor editing is still necessary.

##Specific comments:

Your abstract, especially the methods and results section need more detail. The NMDS is not a method used for clustering, please be more specific. I expect to read a few measures of model quality (for the GLMMs) in the results section. As I guess from your analysis, the relationship between seed productivity and vegetation indices is only good in October, so the quality and the direction of the relationship changes.

For figure 2 the you used a best-case scenario, which I feel means you picked the most distinct plots and pixels for comparison. Thus the generalization that the indices differentiate well between the three clusters needs to be more cautious throughout the text.

The vegetation data (plant cover matrix) for the calculation of the NMDS is still missing, since one of the review points is to check whether all data is available please ask the editor to decide if it is sufficient to submit the article without the plant cover matrix.

L205-209, thank you for adding the sentences, however, they are still difficult to understand to the NMDS as a clustering methods. Since basically you used two methods: clustering with gower distances and wards method? resulting in the colour of the dots in the NMDS diagram, and a NMDS with gower distances. For the NMDS you could also mention the stress level as well as the goodness of fit. Please also state how you decided for the two clusters.

L106: Please reconsider this argument. Including many vegetation indices into a predictive model leads to over fitting. Your models are also only considering a single VI.

L226: Landsat? From the rest of the manuscript I thought you only considered Sentinel 2 imagery.

L237: are proxies of ….
L246: Please also mention the name of the test for spatial autocorrelation in the methods section.

L247: What do you mean by grass count?

L262: Which package did you use for the significance testing.

L 267-273: Please add a few indicators of model quality here. I am not very familiar with GLMM tables and could not figure out on which statistics you base your arguments.

L274: To be more precise you could consider calculating a normal indicator species analysis.

L278: Consider changing quadrats to plots.

L293: vegetation composition, or vegetation cover?

L372: Species name is one time in brackets and the other time without brackets,

Experimental design

no comment

Validity of the findings

no comment

---

## Round 0.3 · Minor Revisions

I was waffling between recommending a minor or major revision. I think some of the recommendations do require some careful consideration and would lead to improvements if acted upon. I agree with Reviewer 1 on the comment mentioned in Experimental design. I invite you to submit a revised version of the manuscript that addresses the points raised during the review process. Please provide either line numbers, the revised sentences themselves, or, preferably, both, in the response to reviewers document.

Reviewer 1 ·

Basic reporting

Major comment

The edits in ll. 152-156, in my opinion, do not clarify how variation in the vegetation type is related to seed productivity, and hence food availability. I suggest that the authors consider moving the explanation they give in ll. 354-357 to the Introduction, and spell out this relationship between grass cover and seed productivity in the Abstract, Results and Discussion as well. I assume this linkage is very obvious to readers familiar with the study site, but I really struggled to understand the relevance of the vegetation cluster analysis for answering the research questions of the study.

Minor comments

Table 2: It is not immediately clear from reading the Methods why the significance of each model was tested by comparing the full model to a model lacking the interaction terms (rather than a null model). Could an explanation be added?
Results section in the Abstract: The results pertaining to the vegetation cluster analysis seem to be missing – could these be added for completeness?
l. 182-187: The authors mentioned they had added an explanation for why Fowlers Gap is a suitable study site, but I cannot find this in the new manuscript – as of now, my original comment (that since the paper does not compare the relative utility of satellite remote sensing and rainfall data for predicting resource availability, or integrate animal population dynamics, I am not sure why this motivated the choice of study area.) still stands.
l. 226: I think “identified by these clusters” should probably be “inside these clusters”.
l. 250: Could you please clarify whether, for the seed productivity analysis, vegetation indices were calculated for every pixel covered by the 50m transects? This is not immediately apparent from the text as it is currently written.
l. 258: Thank you for clarifying this – could the authors additionally clarify how they calculated the average (i.e. was it the mean or median, or the x most productive sites)?
l. 262: I think this should be “product of” instead.
l. 325: I am still unclear about what is meant by “cover variation” – does this mean variation of vegetation cover within in patches or between patches? And variation of what cover (grass, shrubs, …)?

Experimental design

Table 3: I really do believe that the Bonferroni correction is 9, and not 3 (as for each vegetation index the results of 9 separate tests are reported, implying 9 separate tests have been carried out). This means none of the differences are significant, and in my opinion this should be reflected in the Discussion.
Table S3: Similarly, a Bonferroni correction of 3 has to be carried out for each vegetation index, reducing alpha to 0.017 (meaning none of the differences are statistically significant).

Validity of the findings

No comments apart from the issues raised in Experimental Design.

Reviewer 2 ·

Basic reporting

Dear authors,

I believe the manuscript greatly improved and is ready for publication. During the last reading I found a few very minor things, which I ask you to consider:

L52: under which conditions exactly?
Tab. S1 : In vegetation ecology indicator species analysis tables are sorted first by significance, and then by decreasing realative occurence per cluster.

Tab. S3. n=2 seems too low to generate meaningful results in statistical testing for my taste. Please consider to remove it from the paper.

Experimental design

....

Validity of the findings

...

---

## Round 0.4 · accepted · Accept

Thank you for the revised ms. I am happy to accept your paper as is.
Congratulations.